# Psychometric properties and factorial structure of the Spanish version of the HLS-EU-Q16 questionary in Venezuelan adults

Judith Francisco-Pérez[1], Víctor López-Guerra[2*], Gabriel Ortíz[2], Angélica Rojas[3], Carmen Martínez[4], Ronald Serrano[5]

1 Facultad de Enfermería, Grupo de Investigación en Salud Digital, Pontificia Universidad Católica del Ecuador, Quito, Ecuador, 2 Departamento de Psicología, Universidad Técnica Particular de Loja, Loja, Ecuador, 3 Programa de Psicología, Universidad Centroccidental Lisandro Alvarado, Barquisimeto, Venezuela, 4 Carrera de Psicología, Universidad Yacambú, Barquisimeto, Venezuela, 5 Carrera de Enfermería, Universidad Central de Venezuela, Caracas, Venezuela

* vmlopez5@utpl.edu.ec

## Abstract

Health literacy is a variable determined by personal skills and resources for information management and health-related decision making. Studies have shown that it positively influences health maintenance and disease prevention. To contribute to the lack of validated and adapted instruments in Latin America, this study aimed to analyze the psychometric properties of the HLS-EU-Q16 questionnaire short version in Spanish, in Venezuelan adult population. A non-probabilistic intentional sample of 972 (mean age 41.55 years, SD = 15.64; 67.5% female) was surveyed using a cross-sectional design. Data analysis included exploratory (EFA) and confirmatory factor analysis (CFA), factorial invariance, assessment of internal consistency and group contrasting method according to educational level and economic capacity. The three-factor model showed the best fit to the data ($X^2/df$ = 1.52; CFI = .990; TLI = .988; SRMR = .056; RMSEA = .033) and such model remained invariant across sex. The internal consistency was adequate, with Alpha and Omega coefficients for the total scale (α = .88, ω = .88) and three factors: health care (α = .83, ω = .81), disease prevention (α = .73, ω = .73) and health promotion (α = .88, ω = .88). The results indicate that people with postgraduate studies report higher levels of health literacy and people who have borrowed money to buy food or medicines, who have stopped seeing their doctor or taking medicines due to lack of money have low levels of health literacy. The HLS-EU-Q16 questionnaire adapted to Venezuela is reliable, valid and easy to apply. Hence, it will be useful for measuring health literacy and generating preventive programs.

**Data availability statement:** "All relevant data are within the paper and its Supporting information files."

**Funding:** The author(s) received no specific funding for this work.

**Competing interests:** The authors have declared that no competing interests exist.

## Introduction

Several diseases in the Venezuelan population are the result of limitations of the social, economic and cultural context in which people live [1]. Some of these circumstances include a low level of knowledge, difficulties in recognizing some symptoms and a lack of decision-making skills in the face of health conditions [2]. Societal disparities have also been linked to health behaviors and chronic diseases in this country [3].

In recent years, millions of Venezuelans have emigrated in search of better living conditions in other countries of the region. A significant number of them return to Venezuela in worse conditions than those that motivated their departure. These circumstances affect not only health care, but also the self-management that these people can make of their health [4].

In this context, intersectoral actions have been implemented with academic institutions, non-governmental organizations and greater citizen participation. These actions have helped in mitigating the weaknesses of the sector with preventive and health promotion strategies [5].

Health literacy (HL) is the process of acquiring and understanding medical information [6] which determines people's decision making and health management [7,8], whose importance stems from its influence on sanitary behaviors [9] and in the willingness to use health care services [10]. Therefore, HL is essential for disease prevention and treatment as it enables people to interact with the healthcare system and use the information they obtain appropriately.

Previous studies have related HL to the management of chronic diseases [11]. In addition, the caregiving activities and supportive behaviors of caregivers of people with diabetes were greater in those with higher levels of HL [12]. Moreover, further HL limitations have been found in people with financial deprivation [13,14], and low educational levels [15].

Recent research connects HL with preventive and health-promoting behaviors in cancer patients [16,17], and both the adolescents [18], and adult population in the context of COVID-19 [19]. Similarly, its impact self-care and the reduction of perioperative risk factors [20], and cognitive deterioration [21] has been documented.

There are different instruments to measure HL. One of the most widely used scales is the European Health Literacy Questionnaire (HLS-EU-Q) [13]. This is a brief measurement instrument, derived from the broader HLS-EU-Q47 questionnaire, designed by the HLS-EU research consortium. The HLS-EU has 47 items that address self-reported difficulties in accessing, understanding, evaluating, and applying information in tasks related to healthcare decision making, disease prevention, and health promotion [22]. Fig 1 summarizes the key aspects of the model.

The model from Sorensen et al. [22] postulates the key processes of accessing, understanding, evaluating, and applying health-related information within three dimensions: (a) health care (ability to access, understand, interpret, evaluate information about medical or clinical issues, make informed decisions about medical issues, and comply with medical advice); (b) disease prevention (ability to access, understand, interpret, evaluate information about health risk factors, make informed

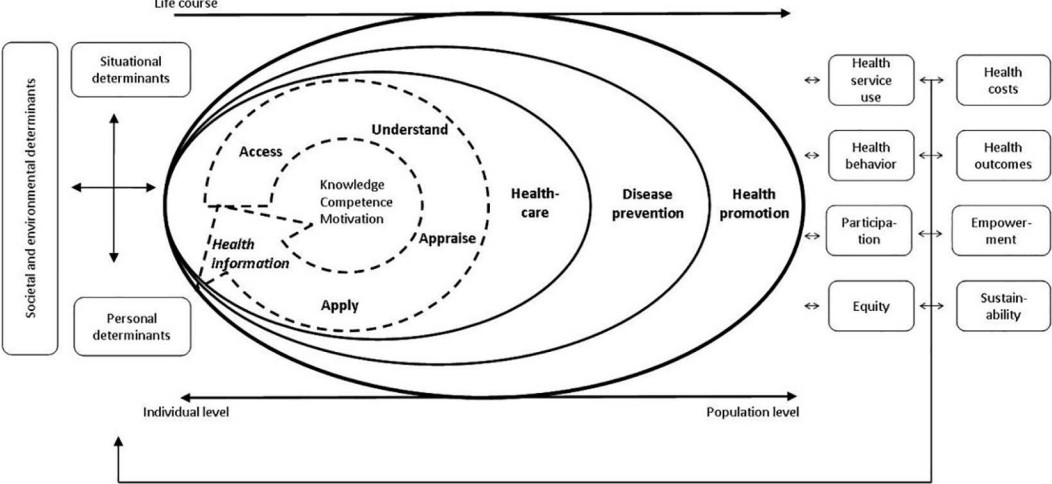

**Fig 1. Integrated model of literacy in health.** *Note: Taken from Sorensen et al. [22].*

decisions about them); and, (c) health promotion (ability to update, understand, interpret, evaluate information about, and make informed decisions about determinants of health in the social and physical environment). Although the HLS-EU-Q47 instrument is considered adequate, a shorter scale of 16 items was created from this version, with better psychometric properties, which minimizes the application time to 3 minutes, is simple to analyze and has a high correlation with the original version [23].

The HLS-EU-Q16 instrument was translated into Spanish and validated in the Valencian region of Spain by Nolasco et al. [24] in 2020, where the study was conducted with 5485 subjects aged 15 or older, and its results indicated that the items were grouped into a single factor that explained 79.1% of the total variability of HL. In that study, all factor loadings exceeded 0.80, indicating that each item contributed significantly to the single factor. The internal consistency of the unidimensional model was very high, with McDonald's omega and Cronbach's alpha values of 0.982, reflecting excellent reliability of the scale.

This HLS-EU-Q16 questionnaire was also validated and adapted in several countries, diverse cultural and linguistic contexts, showing adequate psychometric properties and a high capacity to detect differences in health literacy in different populations: Sweden [8,25], Italy [26] Iceland [27], Romania [28], Portugal [29], France [30]. Within the Latin American context, no validations of the instrument have been identified, except for the one in Mexico [31].

The factorial structure of the questionnaire is diverse. In Spain [24] and Sweden [25] a single-factor internal structure was identified. In contrast, the research conducted in Mexico eliminated 4 items from the original scale, resulting in a two-factor structure: the first focused on actively seeking, understanding and using information provided by health personnel, and the second related to the ability to judge information on self-care and disease prevention [31].

In Portugal, three factors were identified: medical care, disease prevention and health advocacy [29], verifying the original model of Sorensen et al. [22]. However, in a study conducted in Sweden, three factors were also found that do not align with the original model. In this case, the first factor refers to "Finding, understanding and processing information in relation to health," the second refers to "Finding and processing information in connection with health problems," and the third to "Understanding health care information provided by professionals." [8].

Another reference is a research conducted in Iceland that resulted in a four-factor factor structure. The first factor refers to "Processing and use of information from the physician", the second to "Processing and use of information from family

and media", the third to "Processing of information in relation to a healthy lifestyle" and the fourth to "Searching for information about health problems or illnesses [27].

The described background demonstrates that the factor structure of the HLS-EU-Q16 scale has had cross-cultural variations in different contexts. On the other hand, invariance has been little studied in the analysis of instruments to measure health literacy. It is important to include the study of this property in each language version to minimize biases derived from differential item functioning in different groups (e.g., men, women) [30].

Very little research in Latin America has studied the psychometric properties of this questionnaire. Therefore, the aim of this research was to analyze the factor structure, reliability and factorial invariance of the HLS-EU-Q16 short version questionnaire in Spanish language, in a Venezuelan adult population.

Ensuring the psychometric validity of the scale in this new context will help in obtaining accurate and relevant measurements of health literacy, which will facilitate more effective public health interventions. The results obtained in this study will allow us to identify gaps in health literacy and establish the applicability of the HLS-EU-Q16 in the Venezuelan context and in countries where Spanish is the main language. In addition, these results may provide valuable data for the implementation of policies and programs aimed at improving access to and understanding of health information in the adult population.

## Materials and methods

### Participants

The target population of this study consisted of Venezuelan adults aged 18 and older. To define the sample size, a confidence level of 95% and a margin of error of 5% were used. Based on these parameters, the final sample comprised 972 individuals, with an average age of 41.55 years (±15.64), of which 67.5% were women and 32.5% men (see Table 1).

**Table 1. Sociodemographic Characteristics.**

| Variable | M | SD | Min/Max | N | % |
|---|---|---|---|---|---|
| Age | 41.55 | 15.64 | 14/ 84 | | |
| **Gender** | | | | | |
| Male | | | | 316 | 32.5 |
| Female | | | | 656 | 67.5 |
| **Educational level** | | | | | |
| None | | | | 18 | 1.9 |
| Elementary | | | | 71 | 7.3 |
| Secondary | | | | 255 | 26.2 |
| Technical | | | | 99 | 10.2 |
| Graduate | | | | 333 | 34.3 |
| Postgraduate | | | | 196 | 20.2 |
| **Area** | | | | | |
| Rural | | | | 259 | 26.6 |
| Urban | | | | 713 | 73.4 |
| **Chronic illness** | | | | | |
| Yes | | | | 293 | 30.1 |
| No | | | | 679 | 69.9 |
| **Family member with chronic illness** | | | | | |
| Yes | | | | 733 | 75.4 |
| No | | | | 239 | 24.6 |

Participants were selected using a non-probabilistic convenience sampling method, meaning that individuals who were accessible and willing to participate at the time of data collection were included. The data was collected through electronic surveys (Google Forms) and face-to-face interviews conducted in public institutions, parks, or shopping malls. These locations were chosen to ensure a diverse representation of the population, including individuals from different socioeconomic backgrounds and educational levels.

This sampling approach, while non-random, aimed to capture a broad cross-section of the Venezuelan adult population to explore the psychometric properties and factorial structure of the HLS-EU-Q16 questionnaire in this context.

### Instrument

Sociodemographic data: This section included information regarding basic sociodemographic data, such as age, sex, educational level, ethnic self-identification, and information on the status of chronic disease, being a caregiver of a chronic patient, and financial deprivation.

Health literacy: The HLS-EU-Q16 short version in Spanish was used [16], whose original version measures the degree to which people can access, understand, evaluate and use health-related information to make decisions. This is a self-administered instrument consisting of 16 items. Examples of these items include: "How easy is it for you to find information on treatments that your doctor prescribes for your illnesses?", "How easy is it for you to find out where to get professional help when you are ill (e.g., doctor, pharmacist, psychologist)?" and "How easy is it for you to understand what your doctor says to you?" The range of scores for each of the items varies from 1 (very difficult) to 5 (very easy), with high scores indicating the presence of health literacy. The score for each subject was obtained as the sum of the scores for the 16 items, considering an inadequate level (score between 16 and 40), a problematic level (score between 41 and 60) and an adequate or sufficient level (score between 61 and 80). The categorization used in this study aligns with previous HL research. While Sørensen et al. established four levels [32], Nolasco et al. applied a two-level classification [24]. Other studies used a three-level approach [4,15,31] similar to the one adopted in this study. The variability in HL classifications across studies reflects adaptations to specific research objectives and populations. The three-level structure employed here provides a more precise differentiation within the Venezuelan context, aiding in identifying groups with greater intervention needs regarding health information access and comprehension.

### Design and procedure

A cross-sectional study was carried out. Data collection was conducted from October to December 2023. Data collection was performed using electronic forms through google forms and face-to-face surveys outside public institutions, parks or shopping malls.

The study was approved on July 21, 2023, by the Bioethics Committee of the "Universidad Centroccidental Lisandro Alvarado", Venezuela (CBDCS-4–2023), and conducted in accordance with the principles expressed in the Declaration of Helsinki [33]. All participants were provided with written information regarding the study before their participation, in accordance with the ethical guidelines of the Declaration of Helsinki. Therefore, they were aware of the study objectives, the voluntary nature of participation, the potential risks, and the confidentiality of their responses. Data was collected anonymously, and no personal information was recorded. Additionally, no economic compensation was provided for their participation in the study.

The HLS-EU-Q16 is a widely used tool for assessing health literacy in various populations. However, its application in different cultural contexts has required linguistic and cultural adaptations to ensure its validity and reliability. For instance, Gústafsdóttir et al. [27] conducted an Icelandic adaptation that included cognitive interviews, resulting in the reformulation of 11 items to align with the Icelandic culture while maintaining the instrument's conceptual objectives (bmcpublichealth. biomedcentral.com). Similarly, Nolasco et al. [24] noted that some studies have suggested slight modifications to the

original questionnaire to adapt it to different population groups. For example, Storms et al. [34] concluded that, to facilitate its use and interpretation, some items required simplified wording or additional contextual information. In our study, we conducted a linguistic and cultural adaptation of the HLS-EU-Q16 into Venezuelan Spanish, following a translation and back-translation process, complemented by cognitive interviews with 30 participants in a pilot study to ensure comprehension of the questionnaire (see S1 Appendix).

## Data analysis

Statistical analyses were performed with Jeffrey's Amazing Statistic Program (JASP) version 0.18.3.0.

First, a descriptive analysis of the participants' responses to the HLS-EU-Q16 questionnaire was performed by calculating the mean (M) and standard deviation (SD). In addition, significant differences between the means of women and men were evaluated by using an ANOVA.

Secondly, the factor structure was analyzed by performing an exploratory (EFA) and confirmatory (CFA) analysis. Following Harrington's recommendation [35], which suggests performing the EFA and CFA in different samples, the total sample (N = 972) was divided into two independent and homogeneous random subsamples, subsamples nA = 489 and nB = 483. The chi-square test revealed no significant differences between the subsamples, which allowed maintaining a similar proportion of sociodemographic characteristics in both.

The first subsample (nA) was used to perform an AFE to determine the adequacy of the factor loading of each item to the HLS-EU-Q16 questionnaire. Previously, the matrix adaptation was evaluated using the Kaiser-Meyer-Olkin (KMO) method and Bartlett's test of sphericity, where values ≥ .80 for KMO [36] and significance levels $p < .05$ for Bartlett's test [37], indicated the interrelation of the data. The EFA was performed using the principal axis factorization method with oblimin rotation, retaining factor loadings greater than .30 in the rotated matrix [38].

The second subsample (nB) was used to perform the CFA. Considering that the questionnaire is in Likert format and is an ordinal measure, a diagonal weighted least squares (DWLS) estimation method using polychoric correlations was used. This method is recommended for large samples (N > 200) [39]. The indicators selected to evaluate the goodness-of-fit of the models studied were the chi-square ratio (χ2) by degrees of freedom (df), the Bentler comparative fit index (CFI), the Tucker-Lewis index (TLI), the standardized root mean square residual (SRMR), and the root mean square error of approximation (RMSEA). The parameters considered to evaluate the adequacy of the model were: $X^2/df \le 3$ adequate, $\ge 2$ optimal [40]; CFI and TLI ≥ .90 adequate, ≥ .95 optimal; RMSEA and SRMR ≤ .08 adequate, ≤ .05 optimal [41].

In the third place, the factorial invariance of HLS-EU-Q16 was evaluated in the second subsample (nB), considering the following models: configurational invariance (M1), which indicates an unrestricted (baseline) factor structure; metric invariance (M2), which establishes equivalence restrictions between factor loadings; scalar invariance (M3), which includes equivalence restrictions of loadings and intercepts; and strict invariance (M4), which considers equivalence restrictions of factor loadings, intersections and residuals. The measurement invariance and its levels were evaluated according to the recommendations of Cheung and Rensvold [42]: ΔCFI ≤ 0,01 and ΔRMSEA ≤ 0.015.

Fourth, other measures of validity were examined using the contrasted groups method. We analyzed whether two or more groups differed significantly from each other in terms of their variances, by calculating ANOVA for the variables: educational level and economic capacity to buy food or medication, pay for medical consultation and purchase medical treatment, considering $p < .05$ as the criterion for significance. The effect size was also calculated using partial eta squared, where a value of .01 indicates little effect, .06 indicates a medium effect and values greater than .14 indicate a large effect [43]. Prior to the ANOVA, the homogeneity of variances was tested using Levene's test. A detailed analysis was carried out using Tukey's test, to observe whether the means of the groups differed from each other.

Finally, the reliability of the scale was analyzed by internal consistency analysis, evaluating Cronbach's alpha coefficient (α) and McDonald's omega coefficient (ω) considering values ≥ .70 as satisfactory [44].

# Results

## Descriptive analysis

The surveyed individuals (N = 972) have a satisfactory level of health literacy (M = 64; SD = 10.175) according to the HLS-EU-Q16 questionnaire. The scores tend to be located at high values of the scale, with high dispersion around the mean (As = -.0.406; K = -.0.524). In general, the participants in this research show an adequate assessment of health literacy. However, 35% of the sample reports inadequate or problematic levels of this variable.

The normality of the total scale scores was evaluated by means of the skewness and kurtosis coefficients, confirming the fulfillment of this assumption (−1.1) [45]. In addition, we tested whether there were significant differences in health literacy levels between male (M = 63.63; SD = 9.786) and female (M = 64.16; SD = 10.361) study participants. The ANOVA analysis found no statistically significant differences between the two groups (F = .562; gl = 1, 950; p = .454).

## Exploratory factor analysis using subsample nA

A preliminary analysis suggests that the items of the HLS-EU-Q16 questionnaire showed a distribution within the limits of normality. Items have a normal distribution when their skewness is less than 2 and their kurtosis is less than 7 [46]. The analysis showed maximum values of −1.98 for skewness and 3.87 for kurtosis (see Table 2).

The main sample adequacy tests were satisfactory (KMO measurement = .904; Bartlett's test of sphericity: $X^2$ [120] = 7955,965; $p < .001$), therefore, it was considered pertinent to carry out an exploratory factor analysis.

The exploratory factor analysis of the 16 items of the HLS-EU-Q16 questionnaire, performed using the principal axis factoring method with oblimin rotation, yielded a two-factor solution representing 47.8% of the total variance of the test. In the first factor, items 1–6, related to medical care, were grouped, and in the second factor, items 7–16, which assess

**Table 2. Descriptive statistics and factor loadings for HLS-EU-Q16.**

| Items | Mean | SD | Skew | Kurt | Factor 1 | Factor 2 |
|---|---|---|---|---|---|---|
| Item 1 | 3.83 | 1.13 | −0.57 | −0.67 | 0.479 | |
| Item 2 | 3.95 | 1.15 | −0.85 | −0.21 | 0.667 | |
| Item 3 | 4.06 | 1.01 | −1.02 | 0.60 | 0.774 | |
| Item 4 | 4.41 | 0.86 | −1.62 | 2.65 | 0.859 | |
| Item 5 | 3.73 | 1.15 | −0.59 | −0.57 | 0.616 | |
| Item 6 | 3.90 | 1.09 | −0.81 | −0.10 | 0.572 | |
| Item 7 | 4.26 | 0.92 | −1.02 | −0.03 | 0.562 | |
| Item 8 | 3.48 | 1.27 | −0.37 | −1.00 | | 0.399 |
| Item 9 | 4.54 | 0.85 | −1.98 | 3.87 | | 0.494 |
| Item 10 | 4.33 | 1.00 | −1.59 | 1.99 | | 0.374 |
| Item 11 | 3.75 | 1.17 | −0.58 | −0.68 | | 0.735 |
| Item 12 | 3.83 | 1.11 | −0.70 | −0.25 | | 0.796 |
| Item 13 | 4.00 | 1.20 | −0.99 | −0.05 | | 0.472 |
| Item 14 | 4.14 | 0.98 | −1.08 | 0.68 | | 0.467 |
| Item 15 | 4.07 | 0.98 | −1.07 | 0.90 | | 0.805 |
| Item 16 | 4.09 | 1.08 | −1.11 | 0.54 | | 0.597 |
| Percentage of variance | | | | | 24−1% | 23.7% |
| Total variance | | | | | 47.8% | |

Note: Factor 1 (items 1, 2, 3, 4, 5, 6 and 7): Medical care; Factor 2 (items 8, 9, 10, 11, 12,13, 14,15 and 16: Disease prevention and health promotion.
Abbreviations: SD, Standard Deviation; Skew, Skewness; Kurt, Kurtosis.

disease prevention and health promotion, were grouped. All factor loadings were above.30, ranging from.374 to.805, and were statistically significant (p < .001) (see Table 2).

**Confirmatory factor analysis using subsample nB**

To determine the factor structure of the scale, the goodness-of-fit indices of six models of the Spanish version of the health literacy questionnaire were compared (see Table 3):

M1.   One-factor model (items 1 to 16) reported by Nolasco et al. [16] and Mekhail et al. [17].

M2.   Two-factor model eliminating items 1, 5, 8 and 16 reported by García-Vera et al. [31]: items 2, 3, 4, 6, 7, 7, 9, 10 and 13; Factor 2 (Ability to judge self-care and prevention information): items 11, 12, 14 and 15.

M3.   Two-factor model based on the results obtained in the exploratory factor analysis: Factor 1 (Medical care): items 1 to 7; Factor 2 (Disease prevention and health promotion): items 8 to 16.

M4.   Three-factor model proposed theoretically by Sorensen et al [22]: Factor 1 (Medical care): items 1 to 7; Factor 2 (Disease prevention): items 8 to 12; and Factor 3 (Health promotion): items 13 to 16.

M5.   Four-factor model found by Gustafsdottis et al. [27]: Factor 1 items 3, 5, 6 and 7; Factor 2: items 11, 12, 14 and 15; Factor 3: items 4, 9, 10, 13 and 16; and Factor 4: items 1, 2 and 8 and 6.

M6.   Three-factor model proposed by Bergman et al. [8]: Factor 1: items 1, 7, 9, 10, 11, 12, 13, 14, 15 and 16; Factor 2: items 1, 2, 5, 6 and 8; and Factor 3: items 3 and 4.

Based on the adjustment indices, the three-factor M4 (Medical Care, Disease Prevention and Health Promotion) best fits the empirical evidence ($X^2/df$ = 1.52; CFI = .990; TLI = .988; SRMR = .056; RMSEA = .033) compared to models 1, 2, 3, 5 and 6. Additionally, all factor loadings of the items in model 4 were positive and larger than 0.50. The correlations between the first (medical care) and second dimension (disease prevention), between the first (medical care) and third dimension (health promotion) and between the second (disease prevention) and third dimension (health promotion) were high (See Fig 2). This three-factor model is proposed as the final solution for the Venezuelan participants' data and is consistent with the theoretical model of the Sorensen et al. scale [22].

A factorial invariance analysis was performed for the three-factor model 4 of the HLS-EU-Q16 questionnaire, in the second subsample (nB) and by sex, the results of which are presented in Table 4. The configurational invariance (MC) showed indicators of good fit (CFI = 1.000 and RMSEA = .000). Metric invariance (MM) also presented good fit indices (CFI = .999; RMSEA = .011), similar to the MC values, with minimal differences (ΔCFI = .001 and ΔRMSEA = −.011). Factor

**Table 3. Goodness-of-fit indices for the confirmatory factor analysis.**

| Models | CMIN/DF | CFI | TLI | SRMR | RMSEA |
|---|---|---|---|---|---|
| M1 | 3.49 | 0.948 | 0.940 | 0.088 | 0.072 |
| M2 | 3.23 | 0.958 | 0.948 | 0.081 | 0.068 |
| M3 | 1.67 | 0.986 | 0.984 | 0.059 | 0.037 |
| **M4** | **1.52** | **0.990** | **0.988** | **0.056** | **0.033** |
| M5 | 2.16 | 0.977 | 0.972 | 0.069 | 0.049 |
| M6 | 2.09 | 0.978 | 0.974 | 0.067 | 0.049 |

Abbreviations: CMIN/DF, ratio square (χ 2) by degrees of freedom; CFI, Bentler comparative fit index; TLI, Tucker–Lewis index; SRMR, standardized root mean squared residual; RMSEA, Root mean square error of approximation.

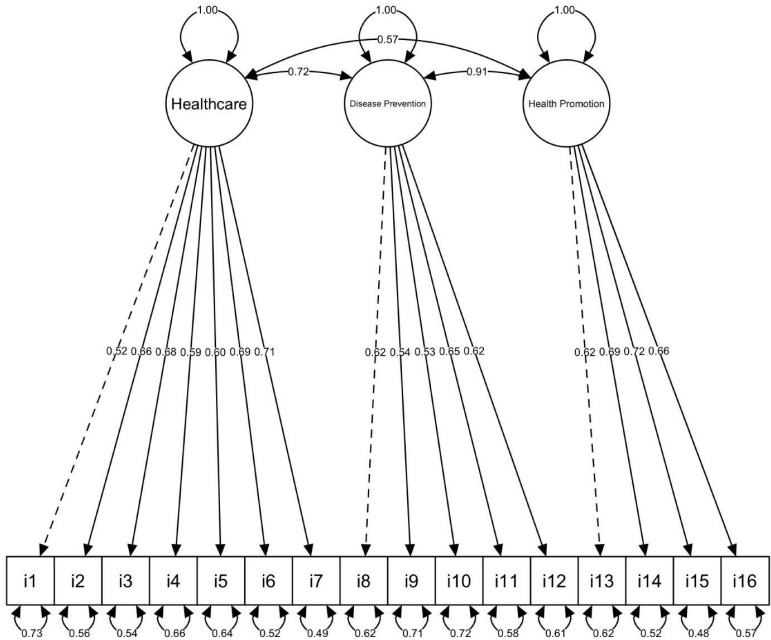

**Fig 2. Diagram of the M4 resulting from the confirmatory factor analysis of the questionnaire HLS-EU-Q16.**

**Table 4. Factorial invariance for the total sample and by gender.**

| Model | $\chi^2$ | df | C-M | $\Delta\chi^2$ | $\Delta df$ | CFI | $\Delta$CFI | RMR | RMSEA | $\Delta$RMSEA |
|---|---|---|---|---|---|---|---|---|---|---|
| **Entire Group** | 153.132 | 101 | – | – | – | **0.990** | – | 0.056 | .033 | – |
| **Men** | 69.691 | 101 | – | – | – | **1.000** | – | 0.065 | .000 | – |
| **Women** | 119.25 | 101 | – | – | – | **0.995** | – | 0.061 | .024 | – |
| MC | 188.944 | 202 | – | – | – | 1.000 | – | 0.062 | .000 | – |
| MM | 220.855 | 215 | MM-MC | 31.91 | −13 | 0.999 | .001 | 0.066 | .011 | −.011 |
| SC | 231.115 | 228 | SC-MC | 10.26 | 13 | 0.999 | .000 | 0.064 | .008 | .003 |
| ST | 241.024 | 244 | ST-SC | 9.909 | 16 | 1.000 | .001 | 0.067 | .000 | −.008 |

Abbreviations χ2, chi-square analyses; df, degrees of freedom; C-M, comparison of factorial invariance models; CFI, Bentler comparative fit index; RMR, standardized root mean squared residual; RMSEA, root mean square error of approximation; Δ, increase. Models: MC, Model Configural; MM, Model Metric; SC, Model Scalar; ST, Model Strict.

loadings did not vary between both sexes, allowing comparison of covariances. Scalar invariance (SC) showed indices equivalent to the previous model (CFI = .999; RMSEA = .008) with no minimal differences (ΔCFI = 0 and ΔRMSEA = .003), accepting invariance between thresholds. Strict invariance (ST) reflected a good fit (CFI = 1.000; RMSEA = .000) with minimal differences (ΔCFI = .001 and ΔRMSEA = −.008), with invariance of the residuals verified. The combined results indicate factorial invariance of SWLS in both sexes.

## Method of contrasting groups with the complete sample

In relation to educational level, significant differences were found between those who have no education and those who report postgraduate studies (master's or doctorate, specialization), with a partial eta squared value of.069 indicating a

moderate effect size. In addition, differences were found between those with primary education and those with postgraduate studies, with a magnitude of .011 indicating little effect.

Regarding economic capacity, statistically significant differences were found between people who borrowed money, sold belongings or spent less to buy food or medicine and those who did not report financial deprivation ($\eta2 = .034$) with a low influence on HL. Differences were also found between those who responded with a "yes" and "no" to the question: *Did you or someone in your family stop going to the doctor because of lack of money?* ($\eta2 = .051$ with a low effect size on HL. Differences were also found between those who responded with a "yes" or "no" to the question: *Did you or someone in your family stop taking any medical treatment because of lack of money?* ($\eta2 = .049$) with a low effect size on HL.

### Internal consistency

The internal consistency for M4 of the HLS-EU-16 questionnaire was satisfactory for both the total instrument ($\alpha = .88$; $\omega = .88$), as well as for its three factors: Health care ($\alpha = .83$; $\omega = .81$), Disease prevention ($\alpha = .73$; $\omega = .73$) and Health promotion ($\alpha = .76$; $\omega = .76$).

### Discussion

The aim of this study was to analyze the psychometric properties of the HLS-EU-Q16 scale in the Venezuelan adult population, due to the scarcity of studies in Latin America and the absence of specific research in Venezuela. Data analysis included procedures of EFA, CFA, factorial invariance, internal consistency assessment and contrasted group methods according to educational level and financial deprivation.

The results reported a tendency for participants to have adequate health literacy, comparable to studies from Iceland [27] and Romania [28] where sufficient levels of health literacy were reported, with 72.5% and 59.2%, respectively. However, they differ from previous studies, in which a high percentage of the sample had problematic levels of HL [9,31].

The high levels of literacy in the Venezuelan sample could be explained by the educational level of the participants, with 26.2% having a high school education and 34.3% having a university education. People lacking formal education had significantly lower levels of health literacy compared to those with postgraduate studies, with a moderate effect size ($\eta² = .069$). Although differences were also found between those with primary education and those with postgraduate studies, the magnitude of the effect was small. The results obtained may also be associated with increased interest in health promotion in the population, as well as the visibility of lifestyles in the collective consciousness [5]. These elements have been associated in previous studies with high levels of health literacy [9,15].

The analysis of the collected data indicates that financial capability significantly influences health literacy. People who faced financial difficulties, such as borrowing money or cutting back on food and medicine, showed lower levels of health literacy. Likewise, those who stopped going to the doctor or taking treatment because of lack of money had lower levels of health literacy. This result is consistent with Sorensen's research [13], which found that groups affected by financial deprivation had higher proportions of people with limited health literacy. In the context of public health, it suggests that public health policies should provide accessible educational resources to vulnerable populations.

No statistically significant differences in health literacy levels were observed between men and women. This finding contrasts with the results of the original study of the instrument in Europe, where gender was found to have a correlation of $r = .05$, indicating that men tend to have slightly lower health literacy [13].

The exploratory factor analysis of the 16 items of the HLS-EU-Q16 questionnaire determined a two-factor structure accounting for 47.8% of the total variance of the test. The first factor is related to medical care and the second factor refers to disease prevention and health promotion. This two-factor adjustment is consistent with the study conducted in Mexico [22], however, it differs in terms of the distribution of the items, since in this research 4 of the 16 items were eliminated, leaving a scale of 12 items.

 

The confirmatory factor analysis proved a better fit in model 4, revealing a three-factor structure. Factor 1 related to Medical Care, Factor 2 to Disease Prevention, and Factor 3 to Health Promotion. These results are in full agreement with the original theoretical model [22] and with the study conducted in Portugal [29]. The research conducted in Sweden [8] had also reported a three-factor structure, however, with an alternative grouping of the items, so that its dimensions were different from the original model.

The results differ from those found in Spain [18], Mexico [22] and Iceland [16], where one-, two- and four-factor factor structures were found, respectively. This demonstrates that the health literacy dimensions of the scale manifest themselves differently across cultures.

Regarding factorial invariance by sex, it was found that the instrument with the three-factor model remained invariant between men and women, demonstrating that the HLS-EU-Q16 is a scale that can be safely applied in male and female Venezuelan populations. Therefore, this ensures that any differences in these groups are due to literacy levels, not to a variation in the scale's performance. This result differs from that found in a study conducted in France [30] where invariance was not maintained, finding differential item functioning according to sex.

When evaluating the internal consistency for model 4 of the HLS-EU-Q16 questionnaire this was shown with levels considered satisfactory both for the total instrument ($\alpha = .88$; $\omega = .88$), as well as for its three factors: medical care ($\alpha = .83$; $\omega = .81$), disease prevention ($\alpha = .73$; $\omega = .73$) and health promotion ($\alpha = .76$; $\omega = .76$). These results are consistent with those obtained in other studies. In Portugal they obtained Cronbach's Alpha values of $\alpha = .89$ overall, $\alpha = .783$ for the health care factor, $\alpha = .724$ for the disease prevention factor, and $\alpha = .703$ for the health promotion factor [29]. In the general scale, results consistent with the study of Mexico ($\alpha = .83$) [31], Iceland ($\alpha = .88$) [27], Romania ($\alpha = .84$) [28] and Francia ($\alpha = .81$) [30] were found.

Several limitations of the study must be mentioned. First, the sample was non-probabilistic, which suggests that future research should include a larger number of people from rural, marginal and vulnerable areas of Venezuela. Second, only the invariance of the measurement between sexes was tested, and it was not possible to evaluate other variables, such as the invariance between different geographic areas (rural or urban) of Venezuela. Third, although data anonymity was guaranteed and the information provided was used for research purposes only, the use of a self-administered questionnaire (such as the HLS-EU-Q16) may be subject to inaccuracies due to social desirability, recall bias and acquiescent responses. Fourth, the sample used in this study is not representative of the 21 countries in which Spanish is the first language, which could affect the external validity of the results. Therefore, for future research, it is recommended to use samples that include participants from different Spanish-speaking countries.

Despite these limitations, the sample size and the values found empirically support our results. In other words, the Spanish version of the HLS-EU-Q16 questionnaire constitutes a reliable and valid instrument that will facilitate health literacy research in Venezuela or in other Spanish-speaking countries, becoming a useful baseline tool for future interventions.

## Conclusiones

The HLS-EU-Q16 questionnaire adapted for the Venezuelan population proved to be a reliable and valid instrument. The three-factor model showed an optimal fit and remained invariant between sexes. Internal consistency was adequate for the total scale and its three factors. In addition, significant differences were found in health literacy related to educational level and financial deprivation. These results indicate that the HLS-EU-Q16 is a useful tool to assess health literacy and can support the development of preventive programs in Venezuela.

## Supporting information

**S1 Appendix. HLS-EU-Q16 Spanish version applied to the Venezuelan adult population.**
(DOCX)

**S1 Data Set. VENEZUELAend.**
(SAV)

## Acknowledgments

We would like to thank the universities where the researchers were based, the participants in the study and the nongovernmental organizations that facilitated access to the application of the instrument.

## Author contributions

**Conceptualization:** Judith Francisco-Pérez, Víctor López-Guerra, Gabriel Ortíz, Angélica Rojas.

**Data curation:** Víctor López-Guerra, Gabriel Ortíz.

**Formal analysis:** Víctor López-Guerra, Gabriel Ortíz.

**Investigation:** Judith Francisco-Pérez, Angélica Rojas, Carmen Martínez, Ronald Serrano.

**Methodology:** Judith Francisco-Pérez, Víctor López-Guerra, Carmen Martínez, Ronald Serrano.

**Supervision:** Judith Francisco-Pérez.

**Validation:** Judith Francisco-Pérez.

**Writing – original draft:** Judith Francisco-Pérez, Víctor López-Guerra, Angélica Rojas.

**Writing – review & editing:** Judith Francisco-Pérez, Víctor López-Guerra, Angélica Rojas.

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
