## [Decision Letter · Decision Letter 0]

Dear Dr. López-Guerra,

1. Strengthen the introduction, including more background on other similar studies.

2. Specify how the population and sample were selected

Please submit your revised manuscript by Mar 22 2025 11:59PM. If you will need more time than this to complete your revisions, please reply to this message or contact the journal office at plosone@plos.org . A rebuttal letter that responds to each point raised by the academic editor and reviewer(s). You should upload this letter as a separate file labeled 'Response to Reviewers'.A marked-up copy of your manuscript that highlights changes made to the original version. You should upload this as a separate file labeled 'Revised Manuscript with Track Changes'.An unmarked version of your revised paper without tracked changes. You should upload this as a separate file labeled 'Manuscript'.

We look forward to receiving your revised manuscript.

Kind regards,

Oriana Rivera-Lozada de Bonilla

Academic Editor

PLOS ONE

2. We note that your Data Availability Statement is currently as follows: [If the data are all contained within the manuscript and/or Supporting Information files, enter the following: All relevant data are within the manuscript and its Supporting Information files]

4, Please review your reference list to ensure that it is complete and correct. If you have cited papers that have been retracted, please include the rationale for doing so in the manuscript text, or remove these references and replace them with relevant current references. Any changes to the reference list should be mentioned in the rebuttal letter that accompanies your revised manuscript. If you need to cite a retracted article, indicate the article’s retracted status in the References list and also include a citation and full reference for the retraction notice.

**Comments to the Author**

1. Is the manuscript technically sound, and do the data support the conclusions?

Reviewer #1: Yes

Reviewer #2: Yes

2. Has the statistical analysis been performed appropriately and rigorously?

Reviewer #1: Yes

Reviewer #2: Yes

3. Have the authors made all data underlying the findings in their manuscript fully available?

Reviewer #1: Yes

Reviewer #2: Yes

4. Is the manuscript presented in an intelligible fashion and written in standard English?

Reviewer #1: Yes

Reviewer #2: Yes

Reviewer #1: This is an interesting review of a Health literacy scale that has been adapted to the local language. Here are a few comments:

Add more literature in the introduction, cite more than one research for the factors thought to be associated with Health literacy (if possible include various socioeconomic locations and global and regional data.)

Line 132 Find a better term for 'Accidental sampling', additionally it's not clear where exactly the sample population was gotten from (at their homes, public areas, schools?)

Generally, can you further explain the difference between assessing the psychometric qualities of the scale VS validating the tool in local context

Reviewer #2: This paper aims to evaluate the psychometric validity of the HLS-EU-Q16 in the Venezuelan context, offering valuable insights into health literacy levels among Venezuelan adults.

A few comments for author's consideration:

1. the adaptation of the HLS-EU-Q16 is mentioned briefly, and many researcher are unaware of this tool. Please expand on its items.

2. Could you provide more details about the informed consent process and how privacy and confidentiality were ensured?

3. Why a factor of 1.5 was used? a reference would be useful.

4. On what basis did you use these thresholds in the scoring system to determine the categories (inadequate, problematic, and adequate)?

5. Why two statistical softwares has been used? please elaborate.

**Do you want your identity to be public for this peer review?** For information about this choice, including consent withdrawal, please see our Privacy Policy

Reviewer #1: No

Reviewer #2: No

---

## [Author Response · Author response to Decision Letter 1]

21 Feb 2025

Revisor 1

1)Add more literature in the introduction, cite more than one research for the factors thought to be associated with Health literacy (if possible include various socioeconomic locations and global and regional data.)

R: References were added that relate LH to chronic disease management, the role of caregivers in diabetes, the influence of education and financial deprivation on LH levels, as well as its impact on prevention and health promotion in cancer patients, adolescents and in the context of COVID-19. Studies linking LH to self-care, perioperative risk reduction and cognitive impairment were also included.

2)Line 132 Find a better term for 'Accidental sampling', additionally it's not clear where exactly the sample population was gotten from (at their homes, public areas, schools?)

R: The term “Accidental sampling” was replaced by “Convenience sampling”, as this term is more accurate and widely accepted in the methodological literature to describe a non-probability sampling based on the accessibility and willingness of the participants.

In addition, it was specified in the manuscript that the data was collected through electronic surveys (Google Forms) and face-to-face interviews in public institutions, parks or shopping malls. These locations were strategically selected to ensure a diverse representation of the population, covering different socioeconomic and educational levels.

3)Generally, can you further explain the difference between assessing the psychometric qualities of the scale VS validating the tool in local context

R:The distinction between the assessment of psychometric properties (reliability, validity and factor structure analysis) and the validation of the tool in the local context (linguistic and cultural adaptation) has been added to the manuscript. This improves the methodological clarity of the study.

Revisor 2

4)The adaptation of the HLS-EU-Q16 is mentioned briefly, and many researcher are unaware of this tool. Please expand on its items.

R:In response to the reviewer’s comment, we have expanded the introduction to detail the linguistic and cultural adaptation of the HLS-EU-Q16. We incorporated references to previous adaptations in different cultural contexts (Gústafsdóttir et al., 2020; Storms et al., 2017) to emphasize the importance of maintaining the instrument’s validity. Additionally, we explicitly describe our process, including translation, back-translation, and cognitive interviews with 30 participants to ensure its comprehension and relevance in the Venezuelan context.

Examples of some items that make up the questionnaire were also provided.

5)Could you provide more details about the informed consent process and how privacy and confidentiality were ensured?

The description of the informed consent process has been expanded to clarify that participants provided written consent, were informed about the study objectives, and assured confidentiality. It is now explicitly stated that data collection was anonymous, and no personally identifiable information was recorded."

6)Why a factor of 1.5 was used? a reference would be useful

R:The use of a factor of 1.5 to account for potential data inaccuracies is not a commonly documented practice in standard psychometric analyses. Therefore, we have removed it from the manuscript to ensure methodological rigor and adherence to established psychometric principles.

7)On what basis did you use these thresholds in the scoring system to determine the categories (inadequate, problematic, and adequate)?

R:The justification for the categorization thresholds has been expanded in the manuscript by incorporating references to previous studies that have used different classification approaches for health literacy.

Sørensen et al. (2015) proposed a four-level classification, combining the ‘inadequate’ and ‘problematic’ levels into a single “limited” literacy category for some analyses.

Nolasco et al. (2020) used a simplified two-level classification, distinguishing between ‘inadequate or problematic’ (0-12 points) and ‘sufficient’ (13-16 points).

Other studies, such as García-Vera et al. (2023), Poza Méndez et al. (2023), and Kampouroglou et al. (2021), adopted a three-level categorization, similar to the one used in this study.

Following these methodological precedents, the present study establishes three levels of health literacy: inadequate (16-40 points), problematic (41-60 points), and adequate (61-80 points). The existence of multiple categorization models across different studies demonstrates that classifications are adapted based on research objectives and target populations. The three-level approach chosen for this study provides a more precise differentiation of health literacy levels within the Venezuelan context, facilitating the identification of groups that require greater intervention in terms of access to and comprehension of health information.

8)Why two statistical softwares has been used? please elaborate.

R:All analyses were performed using Jeffrey’s Amazing Statistics Program (JASP) version 0.18.3.0. Therefore, references to IBM Statistical Package for the Social Sciences (SPSS) version 24 (IBM Inc., Chicago, IL, USA) have been removed to ensure consistency in the description of the statistical software used

Editor

9)Please review your reference list to ensure that it is complete and correct.

R:All references have been carefully reviewed.

---

## [Editor Report · Decision Letter 1]

Psychometric properties and factorial structure of the Spanish version of the HLS-EU-Q16 questionary in Venezuelan adults

PONE-D-24-37744R1

Dear Dr. Victor López-Guerra,

We’re pleased to inform you that your manuscript has been judged scientifically suitable for publication and will be formally accepted for publication once it meets all outstanding technical requirements.

Kind regards,

Oriana Rivera-Lozada de Bonilla

Academic Editor

PLOS ONE

---

## [Editor Report · Acceptance letter]

PONE-D-24-37744R1

PLOS ONE

Dear Dr. López-Guerra,

I'm pleased to inform you that your manuscript has been deemed suitable for publication in PLOS ONE. Congratulations! Your manuscript is now being handed over to our production team.

Kind regards,

on behalf of

Dr. Oriana Rivera-Lozada de Bonilla

Academic Editor

PLOS ONE